# Extended Kalman Filter Design for Tracking Time-of-Flight and Clock Offsets in a Two-Way Ranging System

Sharanya Srinivas, Andrew Herschfelt * and Daniel W. Bliss

Center for Wireless Information Systems and Computational Architectures (WISCA), Arizona State University, Tempe, AZ 85281, USA
* Correspondence: andrew.herschfelt@asu.edu

**Abstract:** As radio frequency (RF) hardware continues to improve, two-way ranging (TWR) has become a viable approach for high-precision ranging applications. The precision of a TWR system is fundamentally limited by estimates of the time offset $T$ between two platforms and the time delay $\tau$ of a signal propagating between them. In previous work, we derived a family of optimal "one-shot" joint delay–offset estimators and demonstrated that they reduce to a system of linear equations under reasonable assumptions. These estimators are simple and computationally efficient but are also susceptible to channel impairments that obstruct one or more measurements. In this work, we formulate an extended Kalman filter (EKF) for this class of estimators that specifically addresses this limitation. Unlike a generic KF approach, the proposed solution specifically integrates the estimation process to minimize the computational complexity. We benchmark the proposed first- and second-order EKF solutions against the existing one-shot estimators in a MATLAB Monte Carlo simulation environment. We demonstrate that the proposed solution achieves comparable estimation performance and, in the case of the second-order solution, reduces the computation time by an order of magnitude.

**Keywords:** wireless sensor networks; two-way ranging; distributed coherence; internet of things; positioning, navigation, and timing; signal processing; spectrum sharing; spectral convergence





## 1. Introduction

As flying, ad-hoc networks (FANETs) become increasingly accessible and affordable, they demand increasing access to the already-crowded frequency spectrum. Intelligent transport systems (ITSs) and internet-of-things (IoT) networks require numerous measurement modalities and are increasingly dissatisfied with the limited reliability and efficiency of legacy wireless sensor network (WSN) solutions. Modern solutions have transitioned towards cooperative and co-design techniques under the category of "spectral convergence" that promise significant performance enhancements and massively increased spectral capacity [1–3].

In a previous work [4], we demonstrated a novel two-way ranging (TWR) system that integrated these modern co-design techniques. Specifically motivated by the limitations of GPS and ZigBee, this system efficiently and simultaneously enabled localization, synchronization, and communication services for wireless unmanned aerial vehicle (UAV) networks. We experimentally demonstrated high-precision ranging (sub-cm) and synchronization (sub-ns) with limited spectral access (10 MHz) and transmit power (1 W) [5] and further addressed the growing vulnerabilities of GPS [6–8] using an integrated encryption and authentication protocol.

While these initial results were successful as a "proof of concept", this early prototype employed a simplified processing chain that is susceptible to outages, cycle slips, shadowing, multipath fading, and numerous other realistic channel conditions. Since FANETs typically require more robust and reliable positioning services, this prototype was largely

considered immature and insufficient for safety-critical applications. To address these limitations, we have developed a family of novel extended Kalman filter (EKF) extensions that enhance and make robust this system. These tracking filters improve the positioning performance by assuming more realistic first- and second-order Markov models and reduce the susceptibility to outages and cycle slips by estimating the parameters over multiple frames rather than the previous "one-shot" approach. We specifically evaluate the proposed methods against our previous results [9] using a simple MATLAB simulation platform and demonstrate notable improvements in performance and processing speed.

### 1.1. System Overview

The proposed two-way ranging system measures the time-of-flight (ToF) between multi-antenna radio platforms, as depicted in Figure 1. The radio platforms operate with independent clocks; so, the measured time-of-arrival (ToA) is affected by both the propagation time $\tau$ and the clock offset $T$ between the platforms. Using each antenna on the platforms, we measure the ToF between each transmit–receive antenna pair, creating a spatial diversity that can then be used to estimate the relative position and orientation of the platforms. We can construct and solve (in closed form) a system of equations using $\tau$ and $T$ and their first derivatives. This closed-form solution is attractive for its computational simplicity and tractability but is sensitive to environmental conditions such as shadowing and fading that might disrupt one or more individual measurements. To mitigate these issues, we propose a family of EKF solutions that achieve comparable or better performance without increasing the computation time; we even demonstrate that the proposed second-order EKF solution reduces the computation time by an order of magnitude compared to its one-shot equivalent.

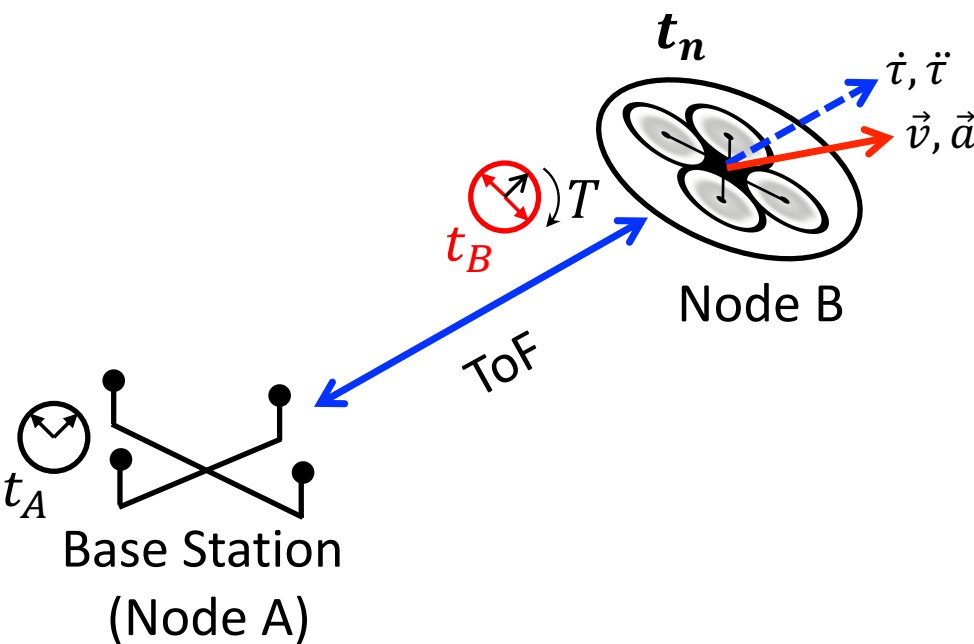

**Figure 1.** Example system configuration of the previously developed two-way ranging (TWR) system that efficiently enables localization, synchronization, and communications in a distributed radio network. We propose a Kalman filter extension to this system that tracks the time offset $T$, time delay $\tau$, and the first and second derivatives thereof between the distributed users. These extensions make the system more robust to cycle slips and reduce the computational complexity by an order of magnitude.

### 1.2. Contributions

In this manuscript, we make the following contributions:

- We derive a family of optimal extended Kalman filter (EKF) tracking algorithms that jointly estimate time-of-flight and clock offsets in a two-way ranging network.
- We implement the proposed EKF solutions and the corresponding optimal one-shot estimators in a simple MATLAB simulation environment.
- We demonstrate that the proposed solution achieves comparable estimation performance to the existing one-shot solutions and, in the case of the second-order solution, reduces the computation time by an order of magnitude.

*1.3. Organization*

The remainder of this manuscript is organized as follows: in Section 2, we briefly review the relevant literature to contextualize our problem and introduce the previous results upon which our proposed solution is built; in Section 3, we specify the problem statement and provide the necessary preliminary definitions; in Section 4, we derive the optimal one-shot estimators for this problem assuming first- and second-order Markov models; in Section 5, we formulate extended Kalman filter tracking algorithms assuming these same models; in Section 6, we benchmark these estimators and algorithms in a simple MATLAB simulation platform; and in Section 7, we provide concluding remarks.

## 2. Background

Robust and precise localization is a critical service for adaptive self-organizing ad hoc networks. Extensive reviews on positioning using the angle-of-arrival (AoA), time-of-arrival (ToA), time-difference-of-arrival (TDoA), and received-signal-strength (RSS) are available in [10–15]. ToA positioning methods [16] are increasingly popular and have received growing interest in the last decade. Multi-antenna systems may leverage ToA measurements to estimate the position and orientation, the precision of which is dictated by the quality of the original measurements [17–19]. For broadcast systems such as the Global Positioning System (GPS) that have extremely high-quality clocks [20,21], it is easy to ignore the timing aspect of this problem; however, for consumer-grade solutions, tracking the relative clock offsets is absolutely critical.

*2.1. Clock Synchronization*

Since TWR-derived positioning relies heavily on precise clock synchronization [22–26], distributed coherence has become a common requisite for these types of networks. Coarse approaches such as the Network Timing Protocol (NTP) are sufficient for communications applications [27–29] but do not support the precision we need for rapid localization. Several extensions have improved this performance [30–36], but many have imposed simplifying assumptions, are not computationally efficient, and lack generality to changes in the underlying models. In response, we proposed a family of novel one-shot estimators in [9] that specifically integrate clock synchronization in the localization algorithm. We demonstrated that these estimators were optimal for single measurements and reduced to a system of linear equations. In this work, we formulate an extended Kalman filter (EKF) extension [37–39] to these estimators to further reduce the computational complexity and improve the robustness to cycle slips and other outages.

*2.2. Statistical Clock Models*

Estimating clock errors involves solving stochastic differential equations whose coefficients are derived from the Allan variance [40] characteristics of the oscillators that drive them [41]. EKF-based clock synchronization is well-studied in this context [42,43]. We draw insights from [44,45] to actualize the joint distributed coherence and ToF tracking using an extended Kalman filter. We do not consider flicker noise while modeling the clocks, and we direct the reader to [46–48] for more information on the subject. Furthermore, we do not consider applications in which the line of sight (LoS) between platforms is obstructed, but this subject is discussed in greater detail in [16,49,50].

## 3. Problem Formulation

In this section, we define the problem statement, summarize the relevant terminology, and establish the underlying assumptions used in the remainder of the document.

### 3.1. Problem Setup

The TWR system presented in [5] enables localization, synchronization, and communications between a network of multi-antenna users. For simplicity, we limit the scope of this discussion to the interactions between just two users labeled $A$ and $B$, as depicted in Figure 1. In this example, user $A$ is a stationary ground station and node $B$ is a UAV moving with some velocity $\vec{v}$ and acceleration $\vec{a}$.

Each transmission takes some time $\tau$ to propagate between these users. We define the relative velocity and acceleration of one node with respect to the other in the direction of the line joining them as radial velocity ($\dot{\tau}$) and acceleration ($\ddot{\tau}$). Users $A$ and $B$ are driven by independent clocks, which at any given time read $t_A$ and $t_B$. The relative time offset $T$ is the time difference between the two clocks, $T = t_A - t_B$. By convention, a positive $T$ denotes that clock $B$ displays an earlier time than clock $A$. The relative frequency offset and drift between these two clocks are denoted $\dot{T}$ and $\ddot{T}$, respectively. These interactions and variable definitions are summarized in Figure 2.

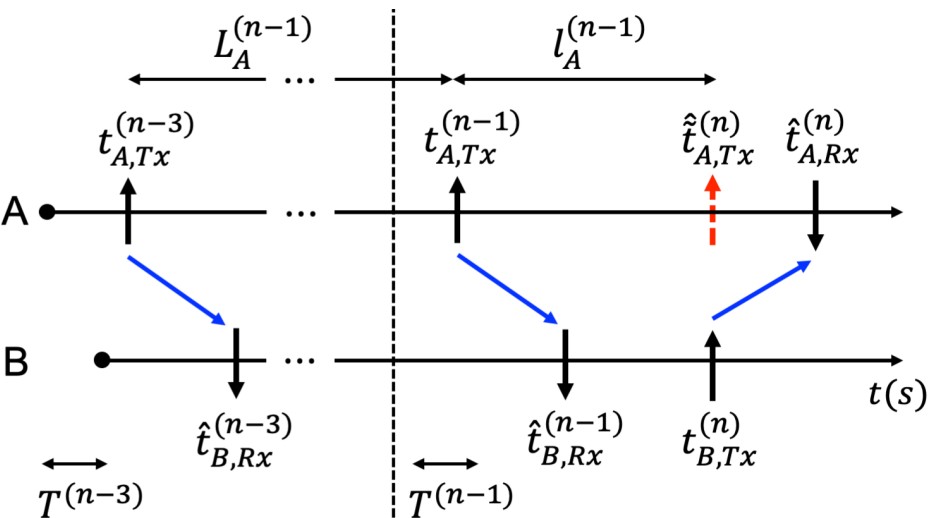

**Figure 2.** Interactions between two users $A$ and $B$. Successive transmissions are separated by $l_A$, and successive cycles are separated by $L_A$. These users exchange timing information each frame to drive the localization algorithm.

### 3.2. Timing Exchange Protocol

Users $A$ and $B$ engage in a cooperative timing exchange in which they alternate transmitting and receiving a waveform that contains both communications feedback and navigation reference sequences. These exchanges are depicted in Figure 2. The transmit timestamps are deterministic and shared, while the receive timestamps are estimated using a variety of ToA estimation techniques. These timestamps are denoted by $t^{(.)}_{(.),(.)}$; the first subscript indicates the node at which the event occurs, the second subscript indicates whether it was a transmit or receive event, and the superscript indicates in which frame the event occurred. Two successive frames comprise a cycle that is $L$ seconds long and is denoted $\{(n-1), (n)\}$ where the successive frames $(n-1)$ and $(n)$ are $l$ seconds apart. Despite rigorous scheduling, the nodes disagree on time, so these cycle and frame lengths are variable and different at each node. These definitions are summarized in the Table 1.

In each cycle, user $A$ transmits a message to $B$ in the frame $(n-1)$, $B$ waits for an agreed frame separation $l$, and transmits a response to $A$ during the next frame $(n)$. Each message contains the transmit and receive timestamps $t_{(.),Tx}, t_{(.),Rx}$ from the previous

frame. This exchange enables estimation of the clock offset $T$ and relative ToF $\tau$ between the two users for each cycle.

**Table 1.** Timing Model Notation.

| | |
|---|---|
| $t_{A,Tx}$ | Transmit event timestamps at node A |
| $t_{A,Rx}$ | Receive event timestamps at node A |
| $t_{B,Tx}$ | Transmit event timestamps at node B |
| $t_{B,Rx}$ | Receive event timestamps at node B |
| $\tilde{t}_{A,Tx}$ | Transmit event time-stamp $t_{B,Tx}^{(.)}$ at node A |
| $l_A$ | Frame length at node A |
| $L_A$ | Cycle length at node A |
| $\tau$ | Relative ToF |
| $\dot{\tau}$ | Relative velocity |
| $\ddot{\tau}$ | Relative acceleration |
| $T$ | Relative time offset |
| $\dot{T}$ | Relative frequency offset |
| $\ddot{T}$ | Relative frequency drift |

For a transmission from $A$ to $B$ during frame $(n-1)$, node $B$ will receive the signal at time $t_{B,Rx}^{(n-1)}$, which is a function of the transmit time $t_{A,Tx}^{(n-1)}$, the time delay $\tau^{(n-1)}$, and the time offset $T^{(n-1)}$, given by:

$$t_{B,Rx}^{(n-1)} = t_{A,Tx}^{(n-1)} + \tau^{(n-1)} - T^{(n-1)}. \tag{1}$$

Similarly, for a transmission from $B$ to $A$ during frame $(n)$, node $A$ will receive the signal at time $t_{A,Rx}^{(n)}$, given by:

$$t_{A,Rx}^{(n)} = t_{B,Tx}^{(n)} + \tau^{(n)} + T^{(n)}. \tag{2}$$

For convenience, we define the transmit timestamp $t_{B,Tx}^{(n)}$ as perceived by node A as $\tilde{t}_{A,Tx}^{(n)}$, which includes the clock offset depicted in Figure 2, as:

$$\tilde{t}_{A,Tx}^{(n)} = t_{B,Tx}^{(n)} + T^{(n)}. \tag{3}$$

At user $A$, the frame length $l$ is perceived as $l_A$, and the cycle separation $L$ is perceived as $L_A$, which for the current cycle becomes:

$$l_A^{(n-1)} = \tilde{t}_{A,Tx}^{(n)} - t_{A,Tx}^{(n-1)}, \tag{4}$$

$$L_A^{(n-1)} = t_{A,Tx}^{(n-1)} - t_{A,Tx}^{(n-3)}. \tag{5}$$

For conciseness, we derive the estimators and tracking algorithms for $T$ and $\tau$ at user $A$. The results are symmetrical and can be applied at user $B$ with trivial substitutions but are excluded for brevity.

### 4. Optimal One-Shot Estimators

In this section, we extend the estimators presented in [19,35] by removing the simplifying assumptions and generalizing the results to higher-order models [9].

### 4.1. First-Order Models

Here, we assume that the relative propagation delay $\tau$ and time offset $T$ between two nodes $A$ and $B$ follow a first-order Markov model. This implies that node $B$ moves with constant radial velocity $\dot{\tau}$ and the clocks on the two radios have constant relative frequency offset $\dot{T}$. During the two-way cycle $\{(n-1),(n)\}$, the delay and offset in frame $(n)$ are expressed as functions of the previous frame $(n-1)$ as

$$\tau^{(n)} = \tau^{(n-1)} + \dot{\tau}^{(n-1)} l_A^{(n-1)}, \tag{6}$$

$$T^{(n)} = T^{(n-1)} + \dot{T}^{(n-1)} l_A^{(n-1)}, \tag{7}$$

where $l_A^{(n-1)}$ is the separation between the two frames. By substituting Equation (1) and some simplifying algebra, we estimate the first-order frame length $l_A^{1,(n-1)}$ as

$$l_A^{1,(n-1)} = \frac{t_{B,Tx}^{(n)} - t_{A,Tx}^{(n-1)} + T^{(n-1)}}{1 - \dot{T}^{(n-1)}}. \tag{8}$$

Using Equations (6)–(8), we can simplify Equations (1) and (2) into a system of linear equations in $\tau^{(n-1)}$ and $T^{(n-1)}$.

$$\tau^{(n-1)} - T^{(n-1)} = \delta^{(n-1)}, \tag{9}$$

$$\varepsilon^{(n-1)} \tau^{(n-1)} + \zeta^{(n-1)} T^{(n-1)} = \eta^{(n-1)}, \tag{10}$$

where $\delta$, $\epsilon$, $\zeta$, and $\eta$ are scalar coefficients defined as:

$$\delta^{(n-1)} = t_{B,Rx}^{(n-1)} - t_{A,Tx}^{(n-1)}, \tag{11}$$

$$\varepsilon^{(n-1)} = L_A^{(n-1)} + t_{B,Tx}^{(n)} - t_{A,Tx}^{(n-1)} + T^{(n-3)}, \tag{12}$$

$$\zeta^{(n-1)} = L_A^{(n-1)} + t_{A,Rx}^{(n)} - t_{A,Tx}^{(n-1)} - \tau^{(n-3)}, \tag{13}$$

$$\eta^{(n-1)} = t_{A,Rx}^{(n)} \left( T^{(n-3)} + L_A^{(n-1)} \right) + t_{B,Tx}^{(n)} \left( \tau^{(n-3)} \right. \tag{14}$$

$$\left. - L_A^{(n-1)} \right) - t_{A,Tx}^{(n-1)} \left( \tau^{(n-3)} + T^{(n-3)} \right).$$

Using Equations (9) and (10), we can construct closed-form estimates of $T$ and $\tau$ at node $A$ for frame $(n-1)$:

$$\tau^{(n-1)} = \frac{\eta^{(n-1)} + \zeta^{(n-1)}\delta^{(n-1)}}{\varepsilon^{(n-1)} + \zeta^{(n-1)}}, \tag{15}$$

$$T^{(n-1)} = \frac{\eta^{(n-1)} - \varepsilon^{(n-1)}\delta^{(n-1)}}{\varepsilon^{(n-1)} + \zeta^{(n-1)}}. \tag{16}$$

Since our two-way ranging configuration is symmetrical, the delay and offset estimators defined in Equations (15) and (16) can be implemented at both nodes $A$ and $B$ with a trivial change of variables or by following the same process outlined in this section with the appropriate substitutions. We therefore exclude the equivalent derivations for node $B$ for brevity for the remainder of the manuscript.

### 4.2. Second-Order Models

The first-order solutions presented in Equations (15) and (16) are closed-form, computationally tractable, and relatively simple to implement. If the underlying behavior of the platform motion or the oscillator imperfections is not well modeled by a first-order Markov model, then this solution may underperform. In some cases, a second-order Markov model may better describe the behavior of these variables; so, we extend the derivation in the previous section to include an additional derivative on $T$ and $\tau$.

Here, we assume that the relative propagation delay $\tau$ and time offset $T$ between two nodes $A$ and $B$ follow a second-order Markov model. During the two-way cycle $\{(n-1), (n)\}$, the delay and offset in frame $(n)$ are expressed as functions of the previous frame $(n-1)$ as

$$\tau^{(n)} = \tau^{(n-1)} + \dot{\tau}^{(n-1)} l_A^{(n-1)} + \frac{1}{2}\ddot{\tau}^{(n-1)} l_A^{(n-1)^2}, \tag{17}$$

$$T^{(n)} = T^{(n-1)} + \dot{T}^{(n-1)} l_A^{(n-1)} + \frac{1}{2}\ddot{T}^{(n-1)} l_A^{(n-1)^2}. \tag{18}$$

Using Equations (17) and (18), we can substitute and simplify Equations (1) and (2) into a system of equations given by:

$$\frac{1}{2}\alpha^{(n-1)} l_A^{(n-1)^2} - \beta^{(n-1)} l_A^{(n-1)} + \gamma^{(n-1)} = 0, \tag{19}$$

where the scalar coefficients $\alpha$, $\beta$, and $\gamma$ are given by:

$$\alpha^{(n-1)} = \ddot{T}^{(n-1)}, \tag{20}$$

$$\beta^{(n-1)} = 1 - \dot{T}^{(n-1)}, \tag{21}$$

$$\gamma^{(n-1)} = T^{(n-1)} + t_{B,Tx}^{(n)} - t_{A,Tx}^{(n-1)}. \tag{22}$$

To solve this system, let

$$\nu^{(n-1)} = \hat{\beta}^{(n-1)^2} - 2\,\alpha^{(n-1)}\gamma^{(n-1)}. \tag{23}$$

A tractable solution to this quadratic equation, for a reasonable set of parameters, is

$$l_A^{2,(n-1)} = \frac{\beta^{(n-1)} - \sqrt{\nu^{(n-1)}}}{\alpha^{(n-1)}}, \tag{24}$$

which exists if and only if $\nu^{(n-1)} \geq 0$. By substituting Equation (24), Equations (17) and (18) reduce to a system of linear equations in $\tau^{(n-1)}$ and $T^{(n-1)}$. It is not convenient to explicitly write out these estimates, so we rely on real-time solvers to perform the estimation.

These first- and second-order estimators are referred to as "one-shot" estimators because they use a single set of measurements to generate a single set of estimates. These one-shot formulations are advantageous because they are relatively straightforward to understand and implement on an actual system, where accessing memory to perform iterative or memory-driven techniques is notoriously expensive. Unfortunately, these formulations are also sensitive to any channel impairments that may corrupt the measurements. This sensitivity can be mitigated by using any number of tracking filter approaches, which typically incur the aforementioned memory access penalties. In the following section, we derive an efficient extended Kalman filter that mitigates this penalty and still retains the near-optimal performance of the original one-shot estimators.

## 5. Extended Kalman Filter Tracking

In this section, we formulate an extended Kalman filter to track the propagation delay $\tau$ and clock offset $T$ between two users $A$ and $B$. The terminology introduced in this section is summarized in Table 2.

**Table 2.** Extended Kalman Filter Notation.

| | |
|---|---|
| $\hat{\mathbf{x}}_{(-)}$ , $\hat{\mathbf{x}}$ | Predicted and estimated state parameters |
| $|\mathbf{x}|$ | Cardinality of state space |
| $\mathbf{F}$ | State transition matrix |
| $\hat{\mathbf{z}}_{(-)}$ , $\hat{\mathbf{z}}$ | Predicted and observed measurements |
| $\mathbf{u}$ | Control parameters |
| $\mathbf{h}$ , $\mathbf{H}$ | Measurement transition and Jacobian |
| $\mathbf{w}$ , $\mathbf{v}$ | State and measurement noise |
| $\mathbf{Q}$ , $\mathbf{R}$ | State and measurement noise covariance matrix |
| $\hat{\mathbf{P}}_{(-)}$ , $\hat{\mathbf{P}}$ | Predicted and estimated state covariance matrix |
| $\mathbf{S}$ | Measurement covariance matrix |
| $\mathbf{K}$ | Kalman gain |

### 5.1. Tracking Preliminaries

In the previous section, we demonstrated that the timestamps from two successive frames $\{(n-1), (n)\}$ contain enough information to estimate $T$ and $\tau$ (see Equations (1) and (2)). Therefore, we employ tracking once every cycle, producing state space estimates at a resolution of 1 cycle = two frames. Kalman filters inherently assume Gaussian processes, which nonlinear problems fails to preserve. Extended Kalman Filters (EKF) [38] counter this issue by linearizing the nonlinear transformations, thus preserving Gaussianity. The algorithm can be visualized in two steps: a) prediction and b) correction. The state space parameters $\hat{\mathbf{x}}$ at time instance $(n-1)$ are predicted using estimates from the previous cycle $\hat{\mathbf{x}}^{(n-3)}$ using the standard transition model given by

$$\hat{\mathbf{x}}_{(-)}^{(n-1)} = \mathbf{F}\left(L_A^{(n-1)}\right)\hat{\mathbf{x}}^{(n-3)} + \mathbf{w}^{(n-1)}, \tag{25}$$

where $\mathbf{F}(\cdot)$ is the state transition, $L_A$ is the cycle length defined in Equation (5), and $\mathbf{w}^{(\cdot)} \sim \mathcal{N}(0, \mathbf{Q}^{(\cdot)})$ is the process noise, assumed to be drawn from a zero-mean multivariate normal distribution. The standard error covariance matrix $\hat{\mathbf{P}}$ in predicting state space parameters can be written as a function of the state transition $\mathbf{F}(\cdot)$ and the process noise covariance matrix $\mathbf{Q}$, given by:

$$\hat{\mathbf{P}}_{(-)}^{(n-1)} = \mathbf{F}^{(n-1)}\hat{\mathbf{P}}^{(n-3)}\mathbf{F}^{(n-1)^T} + \mathbf{Q}^{(n-1)}, \tag{26}$$

where $\hat{\mathbf{P}}^{(n-3)}$ is the covariance matrix from the previous cycle. The standard measurement prediction $\hat{\mathbf{z}}$ is estimated using

$$\hat{\mathbf{z}}_{(-)}^{(n-1)} = \mathbf{u}^{(n-1)} + \mathbf{h}\left(\hat{\mathbf{x}}_{(-)}^{(n-1)}\right) + \mathbf{v}^{(n-1)}, \tag{27}$$

where $\hat{\mathbf{u}}$ represents the control parameters, $\mathbf{h}(\cdot)$ is the nonlinear measurement transition function, and $\mathbf{v}^{(\cdot)} \sim \mathcal{N}(0, \mathbf{R}^{(\cdot)})$ is the measurement noise assumed to be drawn from a zero-mean multivariate normal distribution. We use the measurement Jacobian $\mathbf{H}$, defined as

$$\mathbf{H}^{(n-1)} = \left.\frac{\partial \mathbf{h}(\hat{\mathbf{x}})}{\partial \hat{\mathbf{x}}}\right|_{\hat{\mathbf{x}}_{(-)}^{(n-1)}}, \tag{28}$$

as the *acting* measurement transition function. The error covariance matrix in predicting measurements $\mathbf{S}$ is also typically written as a function of the acting measurement transition function $\mathbf{H}$ and the measurement noise covariance matrix $\mathbf{R}$, which takes the same form as Equation (26):

$$\mathbf{S}^{(n-1)} = \mathbf{H}^{(n-1)}\hat{\mathbf{P}}_{(-)}^{(n-1)}\mathbf{H}^{(n-1)T} + \mathbf{R}^{(n-1)}. \tag{29}$$

Using these predictions, we correct for the state space variables by evaluating a *weighted sum* of the state predictions and deviation in measurement predictions from observations. To do so, we first define a weighting factor $\mathbf{K}$ by normalizing the acting transition function $\mathbf{H}$ by the error covariance matrices in the predicting parameters and measurements, $\hat{\mathbf{P}}$ and $\mathbf{S}$, respectively. This takes the form:

$$\mathbf{K}^{(n-1)} = \hat{\mathbf{P}}_{(-)}^{(n-1)}\mathbf{H}^{(n-1)T}\mathbf{S}^{(n-1)^{-1}}, \tag{30}$$

with which we make the corrections

$$\hat{\mathbf{x}}^{(n-1)} = \hat{\mathbf{x}}_{(-)}^{(n-1)} + \mathbf{K}^{(n-1)}(\mathbf{z}^{(n-1)} - \hat{\mathbf{z}}_{(-)}^{(n-1)}), \tag{31}$$

$$\hat{\mathbf{P}}^{(n-1)} = (\mathbf{I}_{|\mathbf{x}|} - \mathbf{K}^{(n-1)}\mathbf{H}^{(n-1)})\hat{\mathbf{P}}_{(-)}^{(n-1)}, \tag{32}$$

where $|\mathbf{x}|$ is the number of state space parameters. To maintain the resolution of one estimate per frame, we extrapolate between cycles; the estimates $\hat{\mathbf{x}}^{(n)}$ are extrapolated using

$$\hat{\mathbf{x}}^{(n)} = \mathbf{F}\left(l_A^{(n-1)}\right)\hat{\mathbf{x}}^{(n-1)}. \tag{33}$$

Using this construction, we formulate the first- and second-order joint delay–offset tracking algorithm described in Algorithm 1.

---

**Algorithm 1** Extended Kalman Filter Tracking Algorithm

---

1: **while** flight duration **do**

2: $\quad \hat{\mathbf{x}}_{(-)}^{(n-1)} = \mathbf{F}(L_A^{(n-1)})\,\hat{\mathbf{x}}^{(n-3)}$

3: $\quad \hat{\mathbf{P}}_{(-)}^{(n-1)} = \mathbf{F}^{(n-1)}\hat{\mathbf{P}}^{(n-3)}\mathbf{F}^{(n-1)T} + \mathbf{Q}^{(n-1)}$

4: $\quad \hat{\mathbf{z}}_{(-)}^{(n-1)} = \mathbf{u}^{(n-1)} + \mathbf{h}(\hat{\mathbf{x}}_{(-)}^{(n-1)})$

5: $\quad \mathbf{H}^{(n-1)} = \left.\frac{\partial \mathbf{h}(\hat{\mathbf{x}})}{\partial \hat{\mathbf{x}}}\right|_{\hat{\mathbf{x}}_{(-)}^{(n-1)}}$

6: $\quad \mathbf{S}^{(n-1)} = \mathbf{H}^{(n-1)}\hat{\mathbf{P}}_{(-)}^{(n-1)}\mathbf{H}^{(n-1)T} + \mathbf{R}^{(n-1)}$

7: $\quad \hat{\mathbf{z}}_{(-)}^{(n-1)} = \mathbf{u}^{(n-1)} + \mathbf{h}(\hat{\mathbf{x}}_{(-)}^{(n-1)})$

8: $\quad \mathbf{K}^{(n-1)} = \hat{\mathbf{P}}_{(-)}^{(n-1)}\mathbf{H}^{(n-1)T}\mathbf{S}^{(n-1)^{-1}}$

9: $\quad \hat{\mathbf{x}}^{(n-1)} = \hat{\mathbf{x}}_{(-)}^{(n-1)} + \mathbf{K}^{(n-1)}(\mathbf{z}^{(n-1)} - \hat{\mathbf{z}}_{(-)}^{(n-1)})$

10: $\quad \hat{\mathbf{P}}^{(n-1)} = (\mathbf{I}_{|\mathbf{x}|} - \mathbf{K}^{(n-1)}\mathbf{H}^{(n-1)})\hat{\mathbf{P}}_{(-)}^{(n-1)}$

$\quad$ **end**

---

## 5.2. First-Order Extended Kalman Filter

In this section, we assume that the relative propagation delay $\tau$ and time offset $T$ between two nodes $A$ and $B$ follow a first-order Markov model. We model the progression of the state space parameters $\mathbf{x}$ using Equation (25), where the state space variables $\mathbf{x}$ and transition matrix $\mathbf{F}$ at any arbitrary frame are

$$\mathbf{x} = \begin{bmatrix} \tau \\ \dot{\tau} \\ T \\ \dot{T} \end{bmatrix}, \quad \mathbf{F}(l) = \begin{bmatrix} 1 & l & 0 & 0 \\ 0 & 1 & 0 & 0 \\ 0 & 0 & 1 & l \\ 0 & 0 & 0 & 1 \end{bmatrix}, \tag{34}$$

and the process noise covariance matrix is simply modeled as

$$\mathbf{Q}(l) = \mathrm{diag}(\,\mathbf{Q}_\tau(l),\,\mathbf{Q}_T(l)\,), \tag{35}$$

where $\mathbf{Q}_\tau$ and $\mathbf{Q}_T$ are the covariances the in delay and offset transition processes, repespectively (implying that the two are uncorrelated). The delay covariance $\mathbf{Q}_\tau = \mathrm{cov}[\,\tau, \dot{\tau}\,]$ under a Gaussian velocity perturbation model is derived in [44] and is given by:

$$\mathbf{Q}_\tau(l) = \begin{bmatrix} \frac{1}{2}\sigma_{\dot{\tau}}^2 \, l^2 & \sigma_{\dot{\tau}}^2 \, l \\ \sigma_{\dot{\tau}}^2 \, l & \sigma_{\dot{\tau}}^2 \end{bmatrix}, \tag{36}$$

where $\sigma_{\dot{\tau}}^2$ is the variance of the relative velocity and relies on the perturbations induced by environmental attributes such as wind currents. The offset covariance $\mathbf{Q}_T = \mathrm{cov}[\,T, \dot{T}\,]$ is the result of a two-state clock error model, which is derived in [45] and is given by:

$$\mathbf{Q}_T(L_A) = \begin{bmatrix} \sigma_T^2 \, l + \frac{1}{3}\sigma_{\dot{T}}^2 \, l^3 & \frac{1}{2}\sigma_{\dot{T}}^2 \, l^2 \\ \frac{1}{2}\sigma_{\dot{T}}^2 \, l^2 & \sigma_{\dot{T}}^2 \, l \end{bmatrix}, \tag{37}$$

where $\sigma_T^2$ and $\sigma_{\dot{T}}^2$ are the variances in estimating the relative clock and frequency offsets between the two radios, respectively. We previously defined the measurement transition in Equation (27) and now define and substitute the measurements $\mathbf{z}$ and controls $\mathbf{u}$ as:

$$\mathbf{z}^{(n-1)} = \begin{bmatrix} t_{B,Rx}^{(n-1)} \\ t_{A,Rx}^{(n)} \end{bmatrix}, \quad \mathbf{u}^{(n-1)} = \begin{bmatrix} t_{A,Tx}^{(n-1)} \\ t_{B,Tx}^{(n)} \end{bmatrix}. \tag{38}$$

The transition function is formulated by simply substituting the transition model defined in Equations (6) and (7) (the first-order Markov model), which is written as:

$$\mathbf{h}\left(\hat{\mathbf{x}}_{(-)}^{(n-1)}\right) = \begin{bmatrix} \tau^{(n-1)} - T^{(n-1)} \\ \tau^{(n-1)} + \dot{\tau}^{(n-1)} \, l_A^{(n-1)} + T^{(n-1)} + \dot{T}^{(n-1)} \, l_A^{(n-1)} \end{bmatrix}, \tag{39}$$

where the first-order frame length estimate is given by Equation (8), and the measurement covariance matrix can be computed using

$$\mathbf{R} = \mathrm{diag}\left(\sigma_{t_{A,Rx}}^2, \sigma_{t_{B,Rx}}^2\right), \tag{40}$$

where the variances $\sigma_{t_{A,Rx}}^2$ and $\sigma_{t_{B,Rx}}^2$ are the variances of the ToA estimators that generate the measurements [51]. Following the definition presented in Equation (41), the measurement Jacobian may be written explicitly as:

$$\mathbf{H}^{(n-1)} = \begin{bmatrix} \frac{\partial \mathbf{h}(\hat{\mathbf{x}})}{\partial \tau} & \frac{\partial \mathbf{h}(\hat{\mathbf{x}})}{\partial \dot{\tau}} & \frac{\partial \mathbf{h}(\hat{\mathbf{x}})}{\partial T} & \frac{\partial \mathbf{h}(\hat{\mathbf{x}})}{\partial \dot{T}} \end{bmatrix}\Bigg|_{\hat{\mathbf{x}}_{(-)}^{(n-1)}}, \tag{41}$$

where each partial derivative is included below for convenience:

$$\frac{\partial \mathbf{h}(\hat{\mathbf{x}})}{\partial \tau}\bigg|_{\hat{\mathbf{x}}_{(-)}^{(n-1)}} = \begin{bmatrix} 1 \\ 1 \end{bmatrix}, \tag{42}$$

$$\frac{\partial \mathbf{h}(\hat{\mathbf{x}})}{\partial \dot{\tau}}\bigg|_{\hat{\mathbf{x}}_{(-)}^{(n-1)}} = \begin{bmatrix} 0 \\ \dfrac{t_{B,Tx}^{(n)} - t_{A,Tx}^{(n-1)} + T^{(n-1)}}{1 - \dot{T}^{(n-1)}} \end{bmatrix}, \tag{43}$$

$$\frac{\partial \mathbf{h}(\hat{\mathbf{x}})}{\partial T}\bigg|_{\hat{\mathbf{x}}_{(-)}^{(n-1)}} = \begin{bmatrix} -1 \\ \dfrac{1 + \dot{\tau}^{(n-1)}}{1 - \dot{T}^{(n-1)}} \end{bmatrix}, \tag{44}$$

$$\frac{\partial \mathbf{h}(\hat{\mathbf{x}})}{\partial \dot{T}}\bigg|_{\hat{\mathbf{x}}_{(-)}^{(n-1)}} = \begin{bmatrix} 0 \\ \dfrac{(1 + \dot{\tau}^{(n-1)})(t_{B,Tx}^{(n)} - t_{A,Tx}^{(n-1)} + T^{(n-1)})}{(1 - \dot{T}^{(n-1)})^2} \end{bmatrix}. \tag{45}$$

*5.3. Second Order Models*

In this section, we assume that the relative propagation delay $\tau$ and time offset $T$ instead follow the second-order Markov model defined in Equations (17) and (18). The second-order transition matrix has the structure $\mathbf{F}(L_A) = \text{diag}(\, \mathbf{f}(L_A)\,,\, \mathbf{f}(L_A)\,)$, where the state space variables $\mathbf{x}$ and transition matrix $\mathbf{f}(L_A)$ are defined as:

$$\mathbf{x} = \begin{bmatrix} \tau, \ \dot{\tau}, \ \ddot{\tau}, \ T, \ \dot{T}, \ \ddot{T} \end{bmatrix}^\top, \quad \mathbf{f}(l) = \begin{bmatrix} 1 & l & \frac{1}{2}l^2 \\ 0 & 1 & l \\ 0 & 0 & 1 \end{bmatrix}. \tag{46}$$

The delay covariance $\mathbf{Q}_\tau = \text{cov}[\,\tau, \dot{\tau}, \ddot{\tau}\,]$ under a Gaussian acceleration perturbation model is derived in [44] and is given by:

$$\mathbf{Q}_\tau(l) = \begin{bmatrix} \frac{1}{4}\sigma_{\ddot{\tau}}^2 l^4 & \frac{1}{2}\sigma_{\ddot{\tau}}^2 l^3 & \frac{1}{2}\sigma_{\ddot{\tau}}^2 l^2 \\ \frac{1}{2}\sigma_{\ddot{\tau}}^2 l^3 & \frac{1}{2}\sigma_{\ddot{\tau}}^2 l^2 & \sigma_{\ddot{\tau}}^2 l \\ \frac{1}{2}\sigma_{\ddot{\tau}}^2 l^2 & \sigma_{\ddot{\tau}}^2 l & \sigma_{\ddot{\tau}}^2 \end{bmatrix}, \tag{47}$$

where $\sigma_{\ddot{\tau}}^2$ is the variance in relative acceleration. The offset covariance $\mathbf{Q}_T = \text{cov}[\,T, \dot{T}, \ddot{T}\,]$ is a reflection of the clock errors expressed as functions of their Allan variance characteristics. This offset covariance is derived in [41] and is given by:

$$\mathbf{Q}_T(l) = \sigma_T^2 \, q_1(l) + \sigma_{\dot{T}}^2 \, q_2(l) + \sigma_{\ddot{T}}^2 \, q_3(l), \tag{48}$$

$$q_1(l) = \begin{bmatrix} l & 0 & 0 \\ 0 & 0 & 0 \\ 0 & 0 & 0 \end{bmatrix}, \quad q_2(l) = \begin{bmatrix} \frac{1}{3}l^3 & \frac{1}{2}l^2 & 0 \\ \frac{1}{2}l^2 & l & 0 \\ 0 & 0 & 0 \end{bmatrix}, \quad q_3(l) = \begin{bmatrix} \frac{1}{20}l^5 & \frac{1}{8}l^4 & \frac{1}{6}l^3 \\ \frac{1}{8}l^4 & \frac{1}{3}l^3 & \frac{1}{2}l^2 \\ \frac{1}{6}l^3 & \frac{1}{2}l^2 & l \end{bmatrix}. \tag{49}$$

The model parameters $\sigma_T^2$, $\sigma_{\dot{T}}^2$, and $\sigma_{\ddot{T}}^2$ may be rigorously defined as functions of the Allan variances of the oscillators that drive these radios, but this examination is beyond the scope of this report. We refer the reader to [41] for more detailed information.

The measurement prediction model is given in Equation (27), where the measurements **z** and controls **u** remain unchanged from Equation (38), but the transition function is $\mathbf{h}(\cdot)$ now written as

$$
\mathbf{h}\left(\hat{\mathbf{x}}_{(-)}^{(n-1)}\right) = \begin{bmatrix} \tau^{(n-1)} - T^{(n-1)} \\ \tau^{(n-1)} + \dot{\tau}^{(n-1)} \, l_A^{(n-1)} + \ddot{\tau}^{(n-1)} \, l_A^{(n-1)^2} + \\ T^{(n-1)} + \dot{T}^{(n-1)} \, l_A^{(n-1)} + \ddot{T}^{(n-1)} \, l_A^{(n-1)^2} \end{bmatrix}, \tag{50}
$$

where the second-order frame length estimate [9] is given by Equation (24). Following the definition presented in Equation (41), the measurement Jacobian is now given by Equation (51). We compute and simplify these six partial derivatives using Equation (50) and summarize the results in Equations (52)–(57). These derivatives themselves include partial derivatives of $l_A$ with respect to $T$, $\dot{T}$, and $\ddot{T}$, which we can compute using the definition provided in Equation (23). We include the simplified results in Equations (58)–(60) for convenience.

$$
\left.\frac{\partial \mathbf{h}(\hat{\mathbf{x}})}{\partial \hat{\mathbf{x}}}\right|_{\hat{\mathbf{x}}_{(-)}^{(n-1)}} = \left[\frac{\partial \mathbf{h}(\hat{\mathbf{x}})}{\partial \tau} \quad \frac{\partial \mathbf{h}(\hat{\mathbf{x}})}{\partial \dot{\tau}} \quad \frac{\partial \mathbf{h}(\hat{\mathbf{x}})}{\partial \ddot{\tau}} \quad \frac{\partial \mathbf{h}(\hat{\mathbf{x}})}{\partial T} \quad \frac{\partial \mathbf{h}(\hat{\mathbf{x}})}{\partial \dot{T}} \quad \frac{\partial \mathbf{h}(\hat{\mathbf{x}})}{\partial \ddot{T}}\right]\Bigg|_{\hat{\mathbf{x}}_{(-)}^{(n-1)}}, \tag{51}
$$

$$
\left.\frac{\partial \mathbf{h}(\hat{\mathbf{x}})}{\partial \tau}\right|_{\hat{\mathbf{x}}_{(-)}^{(n-1)}} = \begin{bmatrix} 1 \\ 1 \end{bmatrix}, \tag{52}
$$

$$
\left.\frac{\partial \mathbf{h}(\hat{\mathbf{x}})}{\partial \dot{\tau}}\right|_{\hat{\mathbf{x}}_{(-)}^{(n-1)}} = \begin{bmatrix} 0 \\ l_A^{(n-1)} \end{bmatrix}, \tag{53}
$$

$$
\left.\frac{\partial \mathbf{h}(\hat{\mathbf{x}})}{\partial \ddot{\tau}}\right|_{\hat{\mathbf{x}}_{(-)}^{(n-1)}} = \begin{bmatrix} 0 \\ \frac{1}{2} l_A^{(n-1)^2} \end{bmatrix}, \tag{54}
$$

$$
\left.\frac{\partial \mathbf{h}(\hat{\mathbf{x}})}{\partial T}\right|_{\hat{\mathbf{x}}_{(-)}^{(n-1)}} = \begin{bmatrix} -1 \\ 1 + \left(\dot{\tau}^{(n-1)} + \dot{T}^{(n-1)}\right) \left.\frac{\partial l_A(\hat{\mathbf{x}})}{\partial T}\right|_{\hat{\mathbf{x}}_{(-)}^{(n-1)}} + \left(\ddot{\tau}^{(n-1)} + \ddot{T}^{(n-1)}\right) l_A^{(n-1)} \left.\frac{\partial l_A(\hat{\mathbf{x}})}{\partial T}\right|_{\hat{\mathbf{x}}_{(-)}^{(n-1)}} \end{bmatrix}, \tag{55}
$$

$$
\left.\frac{\partial \mathbf{h}(\hat{\mathbf{x}})}{\partial \dot{T}}\right|_{\hat{\mathbf{x}}_{(-)}^{(n-1)}} = \begin{bmatrix} 0 \\ l_A^{(n-1)} + \left(\dot{\tau}^{(n-1)} + \dot{T}^{(n-1)}\right) \left.\frac{\partial l_A(\hat{\mathbf{x}})}{\partial \dot{T}}\right|_{\hat{\mathbf{x}}_{(-)}^{(n-1)}} + \left(\ddot{\tau}^{(n-1)} + \ddot{T}^{(n-1)}\right) l_A^{(n-1)} \left.\frac{\partial l_A(\hat{\mathbf{x}})}{\partial \dot{T}}\right|_{\hat{\mathbf{x}}_{(-)}^{(n-1)}} \end{bmatrix}, \tag{56}
$$

$$
\left.\frac{\partial \mathbf{h}(\hat{\mathbf{x}})}{\partial \ddot{T}}\right|_{\hat{\mathbf{x}}_{(-)}^{(n-1)}} = \begin{bmatrix} 0 \\ \frac{1}{2} l_A^{(n-1)^2} + \left(\dot{\tau}^{(n-1)} + \dot{T}^{(n-1)}\right) \left.\frac{\partial l_A(\hat{\mathbf{x}})}{\partial \ddot{T}}\right|_{\hat{\mathbf{x}}_{(-)}^{(n-1)}} + \left(\ddot{\tau}^{(n-1)} + \ddot{T}^{(n-1)}\right) l_A^{(n-1)} \left.\frac{\partial l_A(\hat{\mathbf{x}})}{\partial \ddot{T}}\right|_{\hat{\mathbf{x}}_{(-)}^{(n-1)}} \end{bmatrix}, \tag{57}
$$

$$
\left.\frac{\partial l_A(\hat{\mathbf{x}})}{\partial T}\right|_{\hat{\mathbf{x}}_{(-)}^{(n-1)}} = \frac{1}{\nu^{(n-1)}} \quad \text{(see Equation (23))}, \tag{58}
$$

$$
\left.\frac{\partial l_A(\hat{\mathbf{x}})}{\partial \dot{T}}\right|_{\hat{\mathbf{x}}_{(-)}^{(n-1)}} = \frac{-1}{\ddot{T}^{(n-1)}} \left(\frac{\dot{T}^{(n-1)} - 1}{\nu^{(n-1)}} + 1\right), \tag{59}
$$

$$
\left.\frac{\partial l_A(\hat{\mathbf{x}})}{\partial \ddot{T}}\right|_{\hat{\mathbf{x}}_{(-)}^{(n-1)}} = \frac{T^{(n-1)} - t_{A,Tx}^{(n-1)} + t_{B,Tx}^{(n-1)}}{\ddot{T}^{(n-1)} \nu^{(n-1)}} + \frac{\nu^{(n-1)} + \dot{T}^{(n-1)} - 1}{\ddot{T}^{(n-1)^2}}. \tag{60}
$$

## 6. Simulation Results

In this section, we describe the creation of a simple MATLAB simulation platform and compare the simulated performance of the optimal one-shot estimators and the proposed extended Kalman filter solutions. We implemented the configuration depicted in Figure 1 and allowed user *B* to follow the 60 s flight trajectory depicted in Figure 3. The users interacted every 50 ms. The transmit timestamps were known and shared, while the receive timestamps were estimated to precision $\sigma = 0.1$ ns, which we have previously demonstrated experimentally [4]. We assumed that the noise covariance matrices **Q** and **R** were known. We considered the real-time adaptive estimation of these measures in an adjacent publication. We implemented the optimal one-shot estimators from [9] to benchmark the proposed extended Kalman filters. We measured the performance using the root-mean-square error (RMSE) on the propagation delay and clock offset. We compare the delay and offset estimation performance in Figures 4 and 5, respectively.

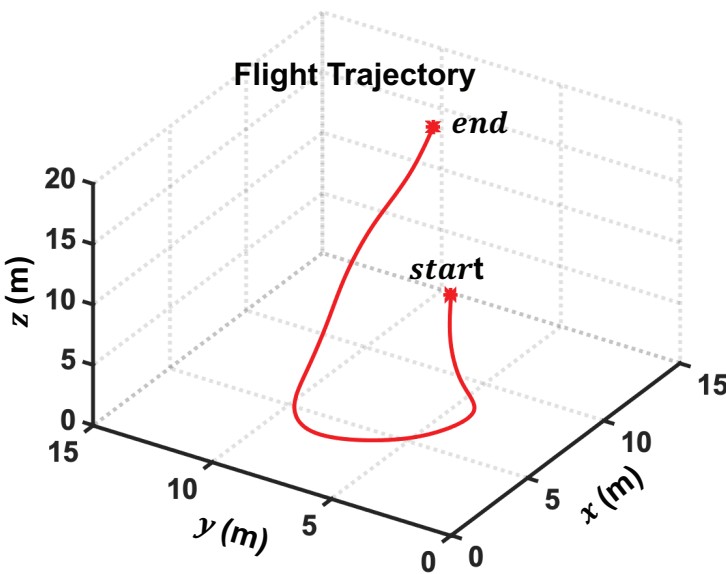

**Figure 3.** Simulated flight trajectory of user *B* for 60 s. User *A* is stationary at the origin.

### 6.1. Simulated Computation Time

We implemented the optimal first- and second-order one-shot estimators defined in [9] and the proposed first- and second-order EKF estimators in MATLAB. We coarsely profiled the average computation time of each of these four solutions in 1000 Monte Carlo simulations over the flight trajectory depicted in Figure 3. MATLAB is a notoriously time-inefficient processing architecture; so, we only used this information to make claims about the relative complexity of each solution, not the actual complexity in an integrated system. These results are summarized in Table 3, which demonstrate that the proposed first-order solution ran slightly slower than the one-shot equivalent, but the proposed second-order solution reduced the run time by over an order of magnitude compared to the second-order one-shot solution.

**Table 3.** Average Estimator Computation Time Across 1000 Trials.

| | |
|---|---|
| Optimal One-Shot Estimator—First-Order | 0.93 ms |
| Optimal One-Shot Estimator—Second-Order | 49.0 ms |
| Extended Kalman Filter Tracking—First-Order | 1.30 ms |
| Extended Kalman Filter Tracking—Second-Order | 2.60 ms |

### 6.2. Simulated Time Delay Estimation Performance

In Figure 4, we plot the time delay estimates $\hat{\tau}$ for the duration of the simulated flight path. In Figure 4a, we plot the true time delay $\tau$ in red and the radial velocity $\dot{\tau}$ in blue. In Figure 4b,c, we plot the estimates generated by the optimal first- and second-order one-shot estimators described in [9], labeled STP1 and STP2, respectively. Both of these estimators successfully estimated the time delay to a precision of about $\sigma = 1$ cm and the radial velocity to a precision of about $\sigma = 15$ cm/s. In Figure 4d,e, we plot the estimates produced by the proposed first- and second-order EKF tracking solutions, labeled EKF1 and EKF2, respectively. The first-order method, depicted in Figure 4d, achieved slightly better performance and converged to a solution after only a single interaction cycle. The second-order method depicted in Figure 4e, however, took several cycles to converge, creating a direct tradeoff between the computational complexity and convergence speed.

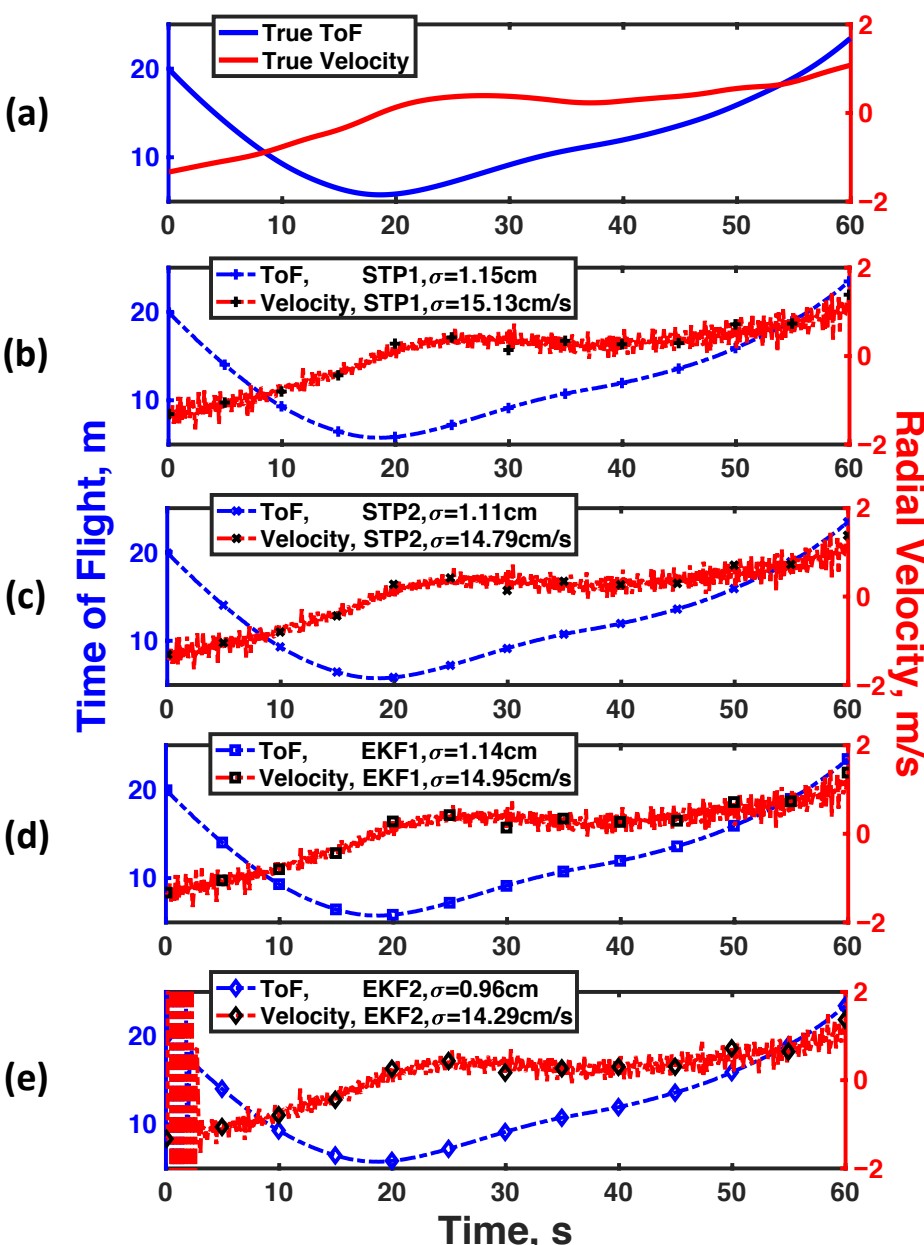

**Figure 4.** (**a**) True time of flight and radial velocity, (**b**,**c**) existing first- and second-order one-shot estimators [9], labeled STP1 and STP2, respectively, and (**d**,**e**) our proposed first- and second-order extended Kalman filter estimators labeled EKF1 and EKF2, respectively.

### 6.3. Simulated Time Offset Estimation Performance

In Figure 5, we plotted the time offset estimates $\hat{T}$ for the duration of the simulated flight path. In Figure 5a, we plotted the true time offset $T$ in red and the clock drift $\dot{T}$ in blue. In Figure 5b,c, we plotted the estimates generated by the optimal first- and second-order one-shot estimators described in [9], labeled STP1 and STP2, respectively. Both of these estimators successfully estimated the time offset to a precision of about $\sigma = 0.05$ ns and the clock drift to a precision of about $\sigma = 1$ ns/s. In Figure 5d,e, we plotted the estimates produced by the proposed first- and second-order EKF tracking solutions, labeled EKF1 and EKF2, respectively. The first-order method depicted in Figure 4d achieved comparable performance and converged to a solution after only a single interaction cycle, while the second-order method depicted in Figure 5e again took several cycles to converge, displaying the same tradeoff between the computational complexity and convergence speed as the previous results.

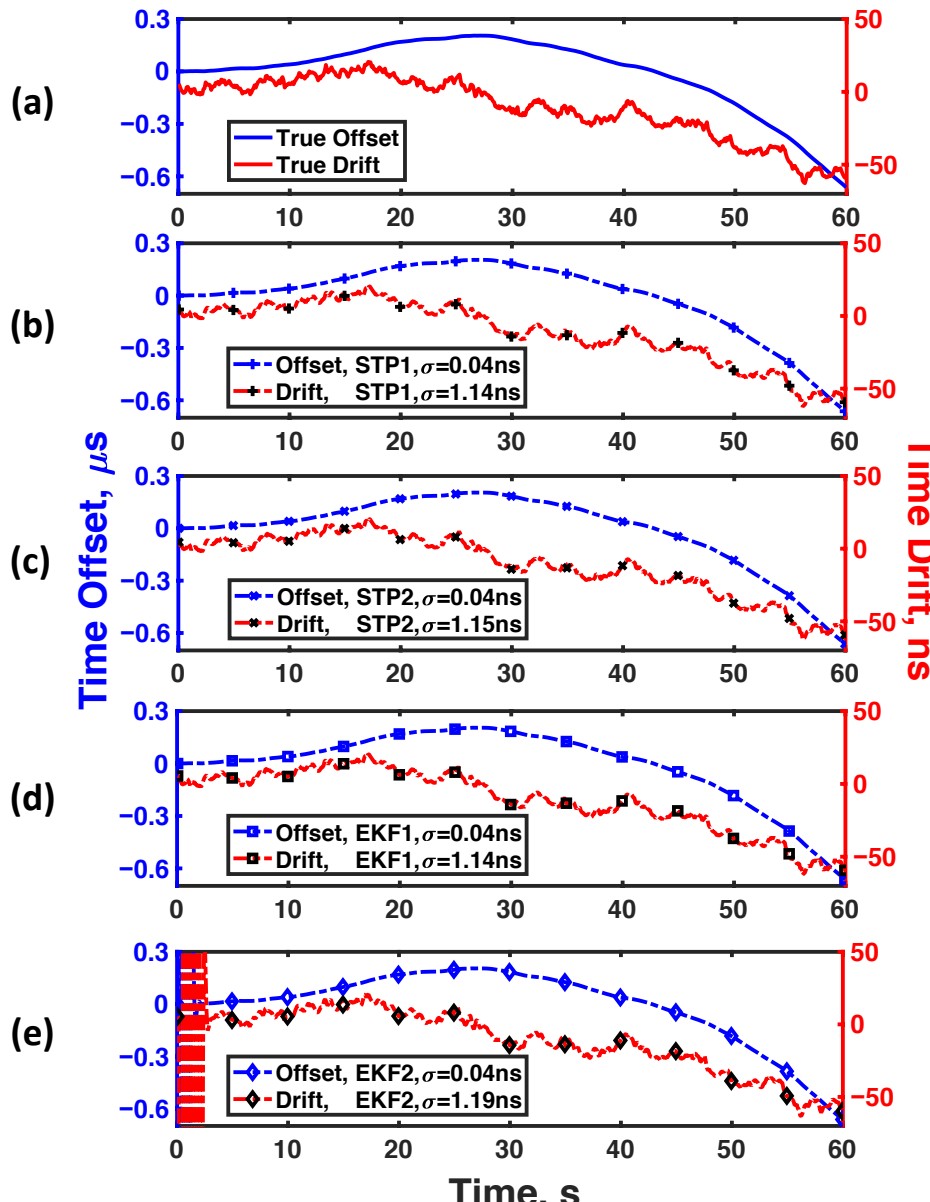

**Figure 5.** (**a**) True time offset and drift, (**b**,**c**) existing first- and second-order one-shot estimators [9] labeled STP1 and STP2, respectively, and (**d**,**e**) our proposed first- and second-order extended Kalman filter estimators labeled EKF1 and EKF2, respectively.

## 7. Conclusions

In this manuscript, we proposed an extended Kalman filter design for tracking time-of-flight (ToF) and clock offsets in a distributed two-way ranging (TWR) network. We proposed this solution to specifically address the sensitivity of the existing one-shot approaches to intermittent channel impairments. Tracking solutions often increase the computational complexity of the estimation processes, which we explicitly mitigate by integrating the estimation process into the EKF solution rather than filtering the outputs of another estimator. We implemented the proposed solutions in a simple MATLAB simulation environment and benchmarked their performance against the corresponding optimal one-shot estimators in the Monte Carlo simulations. We demonstrated that this solution achieves comparable performance to existing, optimal estimators and, in the case of the second-order formulation, actually reduces the computation run time by an order of magnitude.

In the future, we will extend the initialization of the noise covariance matrices to include a real-time adaptive estimation process instead of initializing them using fixed values based on either knowledge of the estimation problem or initial calibration. This is commonly referred to as adaptive extended Kalman filtering (AEKF) and makes the system more robust to wide changes in received signal strength or other channel impairments. In parallel, we are also currently investigating computationally efficient methods for extending these results to higher-order Markov models on the underlying processes.

**Author Contributions:** Conceptualization, S.S. and A.H.; Data curation, S.S.; Formal analysis, S.S. and A.H.; Funding acquisition, D.W.B.; Investigation, S.S. and A.H.; Methodology, S.S. and A.H.; Project administration, D.W.B.; Resources, D.W.B.; Software, S.S.; Supervision, D.W.B.; Validation, S.S. and A.H.; Visualization, S.S., A.H. and D.W.B.; Writing—original draft, S.S. and A.H.; Writing—review and editing, S.S., A.H. and D.W.B. All authors have read and agreed to the published version of the manuscript.

**Funding:** This research received no external funding.

**Data Availability Statement:** The data presented in this study are available on request from the corresponding author. The data are not publicly available due to the policies of Arizona State University regarding pending patents and publications.

**Conflicts of Interest:** The authors declare no conflict of interest.

## Abbreviations

The following abbreviations are used in this manuscript:

| | |
|---|---|
| AoA | angle of arrival |
| EKF | extended Kalman filter |
| FANET | flying ad hoc network |
| GPS | Global Positioning System |
| IoT | Internet of Things |
| ITS | intelligent transport system |
| LoS | line of sight |
| NTP | Network Timing Protocol |
| RMSE | root mean square error |
| RSS | received signal strength |
| TDoA | time difference of arrival |
| ToA | time of arrival |
| ToF | time of flight |
| TWR | two-way ranging |
| UAV | unmanned aerial vehicle |
| WSN | wireless sensor network |

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
