# Peer review of "Extended Kalman Filter Design for Tracking Time-of-Flight and Clock Offsets in a Two-Way Ranging System"

_signals_

Round 1
Reviewer 1 Report
In general the paper contains the required parts of a scientific paper. However, there are some aspects that authors should review.
On the one hand, the introduction section is poor. Authors have mixed introduction and backgorund (which is good), but the problem contextualization is extremely short.
Personally, I prefer to separate both parts, However, it is only a suggestion.
Authors should cite and explain all figures, tables, equations in the text before they appear. For example, Figure 1 is cited 2 pages after the figure position. Please, fix it.
Equations are not cited in the text. Additionally, mathemathical formulation should be better explained. in some cases, it is hard to follow the flow.
Results and graphs should also be better explained.
Finally, conclusion section is too short. Authors should include more details about the obtained results. additionally, it is expected to see some future actions related to the paper proposal.
Reviewer 2 Report
The topic of the paper and the result are interesting. The paper is well-written and easy to read. However, some points must be clarified to be accepted as a publication. Please kindly consider the following suggestions.
- In section 1.2, there are 5 suggested contributions. In its current form, it is only the first contribution is clear enough and can be easily found in the paragraph. However, it is difficult to find the validation of contributions 2-5 in the paragraph. Please give more explanation about the validation of contributions 2-5 in the body of the paper.
- Please give more explanations for the derivation of equation [52]-[60]. In its current form, the explanation is not enough
- In figure 4 and figure 5, the authors compared some results. However, it is not clear which one is the proposed or existing method. For the existing method, please put the citation of references in figure legends. Please also identify the proposed method in the results figure legend.
- Section 5, it lacks an explanation for the results, especially for figures 4 and 5. Please explain the interpretation of the results and the reason why the proposed method is better than the existing method.
- The conclusion is too short and does not give a comprehensive explanation of the research findings. Please give more explanation about the finding of the results, the impact of this research in this field, and the future work of this research.
Reviewer 3 Report
- Please state the main contribution clearly in the abstract.
